# Genome-Wide Association Study of Body Weight Trait in Yaks

**DOI:** 10.3390/ani12141855

**Published:** 2022-07-21

**Authors:** Jiabo Wang, Xiaowei Li, Wei Peng, Jincheng Zhong, Mingfeng Jiang

**Affiliations:** 1Key Laboratory of Qinghai-Tibetan Plateau Animal Genetic Resource Reservation and Utilization, Ministry of Education and Sichuan Province, Southwest Minzu University, Chengdu 610041, China; wangjiaboyifeng@163.com (J.W.); zhongjincheng518@163.com (J.Z.); 2Sichuan Longri Breeding Farm, Hongyuan 610041, China; laoyao1987913@163.com; 3Qinghai Academy of Animal Science and Veterinary Science, Qinghai University, Xining 810000, China; pengwei0112@sohu.com

**Keywords:** yak, genome-wide association study, body weight, heritability, Tibetan Plateau

## Abstract

**Simple Summary:**

The yak is the largest mammal in the Qinghai–Tibetan Plateau and also supplies necessary sources of food and finance for the Tibetan people. Their body weight is a valuable trait for breeding. In the traditional breeding processes, pedigree data were used to evaluate individual genetic potency ability. However, based on the free range husbandry and random mating in domesticated yaks, the genetic improvement in yaks’ body weight is still very slow. In this study, we performed a genome-wide association study using whole genome sequencing data to detect the potential functional genes for body weight in yaks. In total three functional genes were identified as being associated with body weight. The results of this study are important for developing and improving body weight in yaks.

**Abstract:**

The yak is the largest meat-producing mammal around the Tibetan Plateau, and it plays an important role in the economic development and maintenance of the ecological environment throughout much of the Asian highlands. Understanding the genetic components of body weight is key for future improvement in yak breeding; therefore, genome-wide association studies (GWAS) were performed, and the results were used to mine plant and animal genetic resources. We conducted whole genome sequencing on 406 Maiwa yaks at 10 × coverage. Using a multiple loci mixed linear model (MLMM), fixed and random model circulating probability unification (FarmCPU), and Bayesian-information and linkage-disequilibrium iteratively nested keyway (BLINK), we found that a total of 25,000 single-nucleotide polymorphisms (SNPs) were distributed across chromosomes, and seven markers were identified as significantly (*p*-values < 3.91 × 10^−7^) associated with the body weight trait,. Several candidate genes, including *MFSD4*, *LRRC37B*, and *NCAM2*, were identified. This research will help us achieve a better understanding of the genotype–phenotype relationship for body weight.

## 1. Introduction

The Tibetan Plateau is considered the world’s third pole. Yaks (*Bos grunniens*) are the only large, native highland mammals on the Tibetan Plateau and were domesticated over 5000 years ago [1,2]. The body weight (BW) trait is an important selective indicator for the meat production and cold resistance of these animals [3,4].

The BW trait is not only important for the selection and breeding of yaks as meat producers, as larger BWs of yaks correlate with a greater consumption of plateau grassland, which could affect the ecological balance in the Tibetan Plateau, thus significantly impacting the environment [5,6]. The detection of key functional genes will help us to optimize the selection and combination of yaks for economic development and the ecological environment. Based on the male-sterile line between yaks and cattle, traditional crossbreeding methods rarely make enough genetic improvements [7]. The development of molecular tools and genome sequencing technology has gradually facilitated a clearer view of the yak’s genome. The yak reference genome was first sequenced in 2012 [8] and, in 2020, a chromosome-scale yak reference genome was published, which facilitated large population sequencing and genetic deep mining studies in yaks [9].

Genome-wide association studies (GWAS) are widely used to locate candidate genes or markers associated with phenotypes of interest using historical recombination information [10,11,12]. With the development of sequencing technologies and GWAS methods, multiple screening methods have proven to be more effective than single marker testing methods [13]. Jia et al. used the 770 K Illumina Bovine HD Beadchip to associate BW, withers height, body length, and chest girth traits using the multi-locus random-single-nucleotide polymorphism (SNP)-effect mixed linear model (MLM) [14].

In this study, we evaluated BW in 406 Maiwa yak individuals and conducted genotyping by next-generation sequencing at an average coverage of 10×. A total of 25,000 SNP markers were detected and used for evaluation of the population stratification, heritability, identification of association markers, and candidate functional genes controlling BW.

## 2. Materials and Methods

### 2.1. Animals and Phenotyping

The Maiwa yaks used in this study were raised in the Hongyuan prairie (location: Hongyuan, Sichuan, China; geographic coordinates: 102°33′ E, 32°48′ N, altitude: 3570 m) and were obtained from the Longri Breeding Farm of Sichuan Province. This population has undergone selective breeding since the 1980s and has a mortality rate of 5% per year. Currently, the herd is comprised of more than 2000 individuals, ranging in age from 1 to 10 years. In September 2019, the body weight of the majority of the population was measured using a stock weighbridge, and the values were linked to individual earmarks. The collection, breeding, and transportation was in accordance with the Chinese legislation and local stipulation. The protocol for Animal Care and Use was approved by the Animal Ethics and Welfare Association of the Southwest Minzu University (No. 16053), and the experiments were performed according to the regulations and guidelines established by this committee.

### 2.2. Sequencing and SNP Calling

We selected 406 male yaks from whole population that covered all age brackets of yaks. The blood samples of 406 yaks were collected, and DNA was extracted using the TIANamp Blood DNA Kit (Tiangen Biotech Co. Ltd., Beijing, China). After restriction enzyme cut, at least 5 µg whole genome DNA was divided into reads and linked with barcode adapters to construct a sequencing library following the manufacturer’s instructions (Illumina Inc., San Diego, CA, USA). The Illumina Hiseq PE150 (Illumina Inc., San Diego, CA, USA) was used for sequencing at Novogene Bioinformatics Technology Co. Ltd. We utilized a total of 276 GB files downloaded from the sequencing database, and the average Q30 was 88%. The raw data were first quality filtered using fastp software (version 0.19.8) [15] with default parameters. The clean sequencing reads were mapped to the yak reference genome (BosGru 3.0) [16] with BWA-MEM function (version 0.7.17) [17]. SAMtools software [18] was used to convert SAM format files generated by BWA into BAM format and to sort and mark the duplicate reads. Genome Analysis ToolKit (GATK) [19] was used to identify genotype variations containing SNPs and indels. The total gVCF files of all samples were pooled together into a single VCF file, which was converted to a HapMap file using a Perl script coded by ourselves. Markers with more than 10% missing rate were excluded using a filter [20]. All remaining markers were imputed using the Beagle software (version 1.3.2) [21] with default parameters; Beagle uses haplotype clustering strategy and therefore may miss SNPs with a low MAF.

### 2.3. Population Stratification and Heritability Estimation

To identify the population structure in all the accessions, principal component analysis (PCA) was used to indicate the clustered groups and their genetic distance. The Genome Association and Prediction Integrated Tools (GAPIT) [13] software was used to calculate and generate graphics of the PCA with all genotype data, including 25,000 SNP markers and 406 yaks. Here, we estimated the ratio of genotype-explained variance in the total phenotype variance as narrow-sense heritability. As a random effect, the kinship was used to reveal the genetic variance from phenotype. The MLM [22] and compressed MLM (CMLM) [23] in the GAPIT software were used to estimate the variance component. The MLM is a particular type of CMLM that calculates the kinship of each individual, while CMLM estimates an optimum compression level to replace individuals with groups in estimating genetic variance. The cluster method is “mean”; the compress kinship algorithm is “average”. The optimum compression level was confirmed by the model fitting value with negative twice likelihood (−2LL). A much lower −2LL value is better for the fitting model. These two models were integrated into the GAPIT software.

### 2.4. Association Study

In this study, we used three multiple loci testing models, multiple loci mixed model (MLMM) [24], fixed and random model circulating probability unification (FarmCPU) [25], and Bayesian-information and linkage-disequilibrium iteratively nested keyway (BLINK) [26] in the GAPIT package version 3. The multiple loci testing models have been reported to provide more powerful detection than single testing models, such as general linear model (GLM) and mixed linear model (MLM). The Manhattan plots from these methods always show sparse significant signals because the biggest marker effects in an association region were chosen as representative signal in this association region.

The MLMM method can be written as below
(1)$$Y=\mathrm{age}+\mathrm{PCA}+\mathrm{pseudoQTN}+SNP_{i}+K+e$$

Y is the observed phenotype vector. The age is the age bracket indicator matrix for 1 to 10 year old yaks. The PCA is fixed effect to explain the population stratification. The pseudoQTN is significant marker from previous loops that is null when the model begins. SNPi is testing marker in each loop. The K is the kinship between each individuals. The e is residual vector. When the MLMM begins, the model is similar to MLM. After first loop, all significant markers are put into the pseudoQTN, and the MLMM starts with next loop. When the heritability explained by K remains unchanged after loops, the MLMM will stop and calculate final marker effects and *p*-values.

The FarmCPU method can be written as two models
(2)$$Y=\mathrm{age}+\mathrm{PCA}+SNP_{i}+K+e$$
(3)$$Y=\mathrm{age}+\mathrm{PCA}+\mathrm{pseudoQTN}+SNP_{i}+e$$

All parameters are same for MLMM. The function (2) named Random model is used to select pseudoQTN, and the function (3) named Fixed model is used to calculate marker effects and *p*-values. When the likelihood values of Fixed model (3) remain unchanged after loops, the FarmCPU will stop.

The BLINK method is similar to FarmCPU. A different is that the BLINK uses two fixed models, one is used to select pseudoQTN, and the other is used to calculate marker effects and *p*-values. Another difference is that the criterion of model convergence in BLINK is Bayesian information criterion.

The total weights in five continuous years were used as phenotype vector, and the age indicator, PCA, and testing markers were used as fixed parameters in each model. The Bonferroni correction (cutoff = 0.01/N, where N is the total number of markers) was used to avoid Type I error from multiple tests. The GWAS process was carried out following GAPIT default parameters. The Manhattan and quantile–quantile (QQ) plots of results were created with GAPIT software with default code.

## 3. Results

### 3.1. Phenotypic Distribution

A total of 406 records of yaks’ body weight values were tested for normality. As shown in Figure 1, the phenotypic distribution of the whole population presented a normal distribution, with the red line indicating the normal distribution fit. Most of the records ranged between 100 and 235 Kg. The descriptive statistics (mean, variance, maximum, and minimum values) of the BW at different ages are summarized in Appendix A, which indicates that the fourth year is an important criterion to judge Maiwa yaks’ adult weights. After four years of age, variations in the BW of yaks can be primarily attributed to the prevailing environmental conditions. All phenotype values were performed as an outlier analysis using the GAPIT.Remove.outliers function. After removing outliers, outlier values are set to equal the max or min values (Appendix A). Those values more or less than 1.5 times the IQR values were considered as outliers.

### 3.2. Genotype Calling and Marker Density

A total of 2.3 million markers including SNPs, indels, and other variants were detected using the BWA-SAMtools-GATK pipeline with default parameters. After filtering out the SNPs with a missing value >10%, minimum read depth >10, and minor allele frequency >5%, 25,537 SNPs remained. These SNPs were distributed across the 29 autosomes and 2 (X and Y) sex chromosomes (Figure 2A). All individuals had low heterozygosity, approximately 0.01 (Figure 2B), with most SNPs presenting homozygosity (data not shown). Among these SNPs, the average density in the whole yak genome was one SNP/ 110,928 bp (Figure 2C). These results demonstrate that our genetic markers were assigned uniformly across chromosomes and in distinct positions.

### 3.3. Population Structure and Heritability

In this study, our yak population originated from a common farm that was composed of the previous three yak subpopulations from the 1980s. To analyze the population structure of the 406 Maiwa yaks, the whole genome 25,000 SNPs data generated by whole genome sequencing was used for PCA and population stratification. The PCA scatter plot showed a weak population structure for the 406 yaks (Figure 2D). The first three principal components explain more than 3% of the genetic variance. The heritability of BW is estimated by the MLM and CMLM methods (Appendix A). In the MLM method, the genetic variance and heritability were estimated to be 1079.03 and 49.8%, respectively, while in the CMLM, the values were estimated to be 936.5 and 39.5%, respectively. Comparing the −2LL of the two models, CMLM (4119.01) was found to be better than MLM (4120.31).

### 3.4. GWAS and Candidate Genes

To identify SNP markers associated with BW and compare results of different GWAS methods, 406 individual yak genotypic and phenotypic data were analyzed using MLMM, FarmCPU, and BLINK, all of which had been integrated into GAPIT3 software. A total of seven SNPs passed the 1% threshold line after a Bonferroni correction (*p*-values < 3.91 × 10^−7^) and were associated with BW (Figure 3A). Among them, one marker, rs13727 in chromosome 3, located at 28,379,856 bp, was detected by all three GWAS methods. Two markers, rs13559 in chromosome 1, located at 25,773,622 bp, and rs371363 in chromosome 7, located at 87,945,452 bp, were detected by two GWAS methods. Other markers were detected by only one GWAS method. Within up and downstream 100 Kb linkage-disequilibrium (LD) interval information, significant SNPs were associated with candidate genes (Table 1). The QQ plot results of markers–trait associations for BW were shown using the observed against expected *p*-values from three GWAS methods (Figure 3B). A BW difference between three significant SNP genotypes were observed in the different subpopulations (Figure 4). No heterozygous genotype was detected for these three SNPs, and only a single individual in the high subpopulation possessed rs13559 and rs137207 markers.

## 4. Discussion

Yaks usually mature later than cattle and buffalo, often taking more than four years to reach adulthood and a relatively stable body weight and size. The major factor affecting BW prior to reaching maturity is age, but another important factor is the variety of yak. The largest domestic yak is the Jiulong, in which, adult males can reach a body size over 160 cm in length, 130 cm high, and a 205 cm chest circumference, with a BW up to 540 Kg. The Maiwa yaks are the second largest domestic yaks in the world, with a male adult BW reaching up to 320 Kg [27,28]. Random mating occurs within the farming population of most domestic yaks; therefore, it is likely that the coefficient of inbreeding in the subpopulation studied will be significantly higher than that of the general population. Based on the similar growing environment, BW shows high heritability in the subpopulation (Appendix A).

The yak is a different species of the *Bos* genus. Although there are 30 pairs of chromosomes in yaks’ genomes, male offspring hybridized from yak and cattle are infertile. The variance in BW is mainly based on outbreeding from different yak breeds, and the genetic exchange of key genes always exists in common regions associated with BW. The genetic mechanism of yaks also shows differences between cattle and buffalo. Based on the GWAS results, the genomic predicted accuracy of BW, body size, and other traits is far lower than that found in cattle [29]. Many anticipated candidate genes were not detected in the GWAS results, indicating that alternative analyses, such as pathway analysis, are needed to identify whether genes associated with these candidates are detected. The Maiwa yaks’ populations are the second largest for a domestic breed in the Qinghai–Tibet Plateau. The PCA results revealed three major subpopulations, indicating that the genetic differentiation among all Maiwa yaks is relatively high.

Based on the up and downstream 100 Kb LD interval information, the SNP rs137207 (chr3-28,379,856) was annotated by Ensembl, with candidate genes containing major facilitator superfamily domain-containing 14A (*MFSD14A*), SAS-6 centriolar assembly protein (*SASS6*), tRNA methyltransferase 13 homolog (*TRMT13*), leucine-rich repeat-containing 39 (*LRRC39*), and dihydrolipoamide branched chain transacylase E2 (*DBT*). In this region, all these genes were reported by Liu et al. as being associated with milk production traits in water buffalo [30]. One of the *MFSD* genes family, *MFSD4,* was consistent for the main intake effect in skeletal muscle [30]. The *MFSD14A* gene, also called *Hiat1*, is a protein-coding gene with the product of a transmembrane protein with homology to a solute carrier protein family [31]. Based on the Inferred from Electric Annotation (IEA), the *MFSD14A* was annotated with the transmembrane transporter activity Gene Ontology (GO) term. One of the *LRRC* gene’s family, *LRRC37B,* was consistent for body size in pigs [32]. In the IEA database, the GO molecular term of the *NCAM2* gene is protein binding.

The SNP rs371363 (chr7-87,945,452) was annotated with U1 spliceosomal RNA, indicating that the gene type is an snRNA. The SNP rs10942 (chr1-22,314,135) was annotated with neural cell adhesion molecule 2 (*NCAM2*), which has been proved to be associated with body weight in Simmental cattle [33]. Another study revealed that the *NCAM2* gene is associated with body size in ducks [34]. In the IEA database, the GO molecular term of the *NCAM2* gene is protein binding and identical protein binding. This evidence will give us a better understanding of the function of the *NCAM2* gene. The SNP rs118493 (chr2-140,542,613) was only annotated with the tissue factor pathway inhibitor (*TFPI*). In 2017, a genome study using 76 Mongolian yaks’ sequencing data indicated *TFPI* and another 210 genes were located in the introgression segments [2].

In conclusion, we conducted a GWAS analysis of BW in Maiwa yaks using whole genome sequencing with three multiple steps methods. Seven SNPs were shown to be associated with BW, and several candidate genes, including *MFSD4, LRRC37B*, and *NCAM2*, were implicated in the BW phenotype. This research will help us achieve a better understanding of the relationship between the genotype and the BW phenotype. The results could serve as basic information for quantitative trait locus mapping or candidate gene cloning to understand the mechanism of BW in yaks.

## Figures and Tables

**Figure 1 animals-12-01855-f001:**
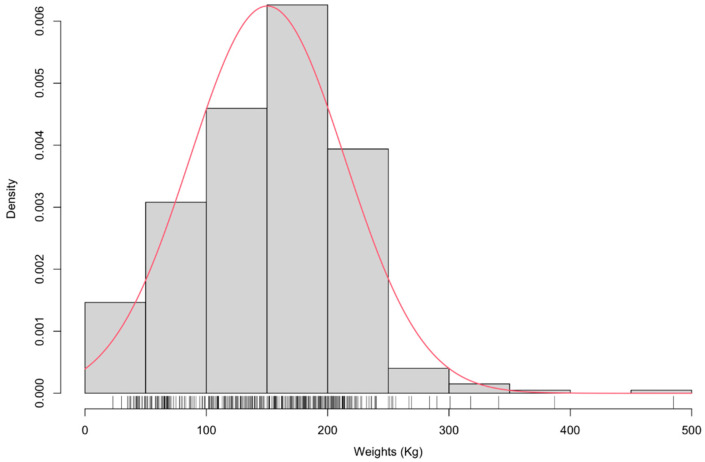
The distribution of body weights in 2019. In total 406 weights of individuals yaks were observed from 23 to 485 Kg. The density of phenotype values was drawn as vertical bar. The red line fits normal distribution.

**Figure 2 animals-12-01855-f002:**
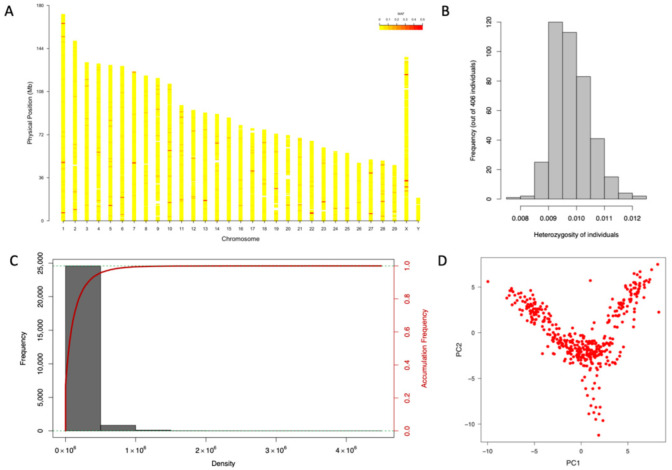
The distribution of genotype and population stratification. In total 25,537 SNPs are marked in the 29 autosomes and two (X and Y) sex chromosomes. The relative positions in the chromosomes were used to indicate marker density (**A**). The heterozygosity frequency of all 406 individuals is shown as bar plot (**B**). Marker density and accumulation frequency are plotted in the figure (**C**). All markers were used to interpret population structure with PCA (**D**).

**Figure 3 animals-12-01855-f003:**
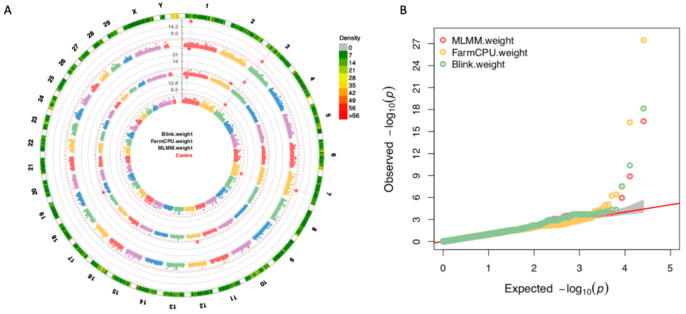
Manhattan plots with all genotypes with three GWAS methods. The GWAS results of three GWAS methods were integrated into circle multiple Manhattan plots (**A**). The outer ring is the marker density and the significant markers were marked as red star. The markers detected by more than two methods were drawn with gray string line. The QQ plots of multiple methods were also integrated into figure (**B**).

**Figure 4 animals-12-01855-f004:**
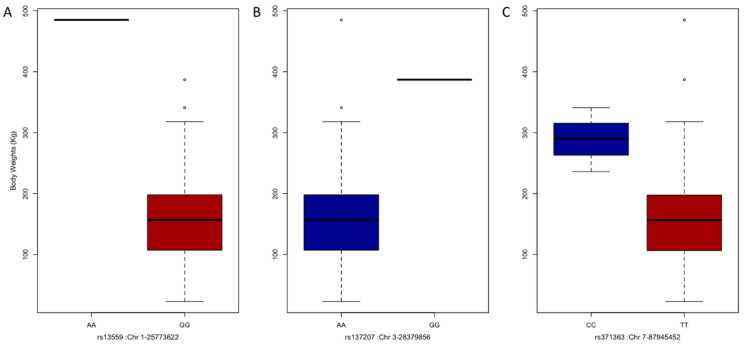
Phenotype distribution among the genotypes of the associated SNPs. The population was divided with the genotype of associated SNPs containing rs13559 (**A**), rs137207 (**B**), and rs371363 (**C**). In the figure there is only one individual with AA or GG genotype for SNP rs13559 or rs137207. For these three associated SNPs, there are no heterozygous genotypes observed in all individuals.

**Table 1 animals-12-01855-t001:** The candidate genes’ information of associated significant SNPs in the 200 kb region. The SNP No. indicates order number in the whole marker list. The chromosome and position mean the physical location information in the genome data. The gene names were annotated from GTF file in the BosGru_v3.0 reference genome. The values in the brackets are distance of base pair between such marker and nearest gene. Class indicates the type of gene, and the region transcribed shows the region SNPs located in.

SNP No.	Chromosome	Position (bp)	Gene Name
rs13559	1	25,773,622	ENSBGRG00000000052-ENSBGRG00000000053
rs137207	3	28,379,856	*MFSD14A*, *SASS6*, *TRMT13*, *LRRC39*, *DBT*
rs371363	7	87,945,452	*U1*
rs10942	1	22,314,135	*NCAM2*
rs118493	2	140,542,613	*TFPI*
rs815163	19	16,238,800	*BPTF*, *KPNA2*

## Data Availability

There are 406 Maiwa yaks Bioprojects accessible at NCBI Bioproject (http://www.ncbi.nlm.nih.gov/bioproject, accessed on 1 May 2022) under accession numbers of PRJNA818054.

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
