# Peer review of "Genome-Wide Association Study of Body Weight Trait in Yaks"

_animals, 2022, doi:10.3390/ani12141855_

Round 1

Reviewer 1 Report

Great work, expensive and well done!

The introduction provides all important information of the general problem that the article addresses. The methodology section is sufficiently detailed and the study design and statistical analysis methods are appropriate, given the study purpose. The results section is logical presented following the analysis plan described in the methods section. All conclusions are supported by the results. In conclusion, experiments have been conducted rigorously. The manuscript is presented in an intelligible fashion, technically sound, and the data support the conclusions. 

Comments:

Please provide a reference (to 2.2. Sequencing and SNP calling) for filters of markers with more than 10% missing rate.

Some observations:

Manhattan plot (figure no. 3) have to be clear much clear (increase the pixels density). The same observation for figure no. 2.

Author Response

Great work, expensive and well done!

The introduction provides all important information of the general problem that the article addresses. The methodology section is sufficiently detailed and the study design and statistical analysis methods are appropriate, given the study purpose. The results section is logical presented following the analysis plan described in the methods section. All conclusions are supported by the results. In conclusion, experiments have been conducted rigorously. The manuscript is presented in an intelligible fashion, technically sound, and the data support the conclusions. 

 [Respond] Thanks for your praise. We will follow your comments and continue our work to consummate this manuscript.

Comments:

Please provide a reference (to 2.2. Sequencing and SNP calling) for filters of markers with more than 10% missing rate.

 [Respond] Following your comments, we have added a reference.

Some observations:

Manhattan plot (figure no. 3) have to be clear much clear (increase the pixels density). The same observation for figure no. 2.

[Respond] Thanks for your comments. I have supplemented separate high pixels figures.

Reviewer 2 Report

Wang et al. performed the GWAS for body weight using 406 Maiwa yaks with the whole genome sequencing data at 10X coverage. The authors identified seven markers for body weight. The sample size for GWAS is small and not ideal. There are several issues with the current analyses.

First, the authors should not randomly select animals for genotyping as the body weight is varied by age of animals. In addition, figure 1 showed some potential outliners in the data, but the authors did not perform any quality control for removing them.

My second concern is about the GWAS model. The authors did not account for any confounding effects in the models such as sex, age, batch, and relationship of animals (sire and dam).

Thirdly, with a low-quality control for filtering SNPs, mentioned in lines 161-163, it is pretty surprising that the authors have only 25,000 SNPs. Moreover, it is not clear about 2.3 million markers including indels, or not. Some information from methods are results are not clear, when did the authors impute the missing genotyping, and what are the parameters the authors use for imputation.

Minors:

The author used only bodyweight as a phenotype in the study so the authors should not use traits or traits after bodyweight

Why did the authors include sex chromosomes in the GWAS? How many yaks are males and females?

The quality of figures is low, and the quality of writing is low as well.

The reference style is not following the journal style.

The authors should have a simple summary and conflict of interest statement.

The threshold for Bonferroni should be 0.05/N

Author Response

Wang et al. performed the GWAS for body weight using 406 Maiwa yaks with the whole genome sequencing data at 10X coverage. The authors identified seven markers for body weight. The sample size for GWAS is small and not ideal. There are several issues with the current analyses.

[Respond] Thanks for your reviewed our manuscript. Your comments are very useful and important to improve our manuscript revised for understood by wide users and readers.

First, the authors should not randomly select animals for genotyping as the body weight is varied by age of animals. In addition, figure 1 showed some potential outliners in the data, but the authors did not perform any quality control for removing them.

[Respond] Thanks for your comments. We used the GAPIT to perform GWAS. In the GAPIT, there is a function named GAPIT.Remove.outliers, that will make outliers as max or min values. We have supplemented a Figure S2 to show the details and distribution of phenotype after that function. And we also supplemented words to descript this program. Based on the free range husbandry of yaks, the determination of weight and genotyping barely covered the most individuals of whole population. So we called that random selected for covered all age brackets of yaks. We have revised these details to make those sentences easily understood.

My second concern is about the GWAS model. The authors did not account for any confounding effects in the models such as sex, age, batch, and relationship of animals (sire and dam).

[Respond] We agree with your comments, that is our mistake of not clearly explanation of model details in the GWAS. All sampled yaks are males in the same field, so that there are genotype markers in the Y chromosome. The ages were treated as fixed effect in the model, the pedigree message were not used in this study. I have supplemented all these details in the Materials.

Thirdly, with a low-quality control for filtering SNPs, mentioned in lines 161-163, it is pretty surprising that the authors have only 25,000 SNPs. Moreover, it is not clear about 2.3 million markers including indels, or not. Some information from methods are results are not clear, when did the authors impute the missing genotyping, and what are the parameters the authors use for imputation.

[Respond] Thanks for your comments. 2.3 Million markers contained all SNP, Indel, and other variants. 25K SNPs were kept that only occurred in 1.2% of total markers. That is because of the high missing rate of genotype in the sequencing results among the whole population. We guess that may be caused by incomplete reference genome sequencing. So we used 10% as a missing cutoff to filter high missing markers and used Beagle to impute genotype values. We followed your comments and supplemented the necessary details of imputation.

Minors:

The author used only bodyweight as a phenotype in the study so the authors should not use traits or traits after bodyweight

[Respond] We agree with your comments and have revised relative sentences.

Why did the authors include sex chromosomes in the GWAS? How many yaks are males and females?

[Respond] All yaks are males. We have supplemented gender information in the Materials.

The quality of figures is low, and the quality of writing is low as well.

[Respond] We have re-created all figures with high pixels, and revised sentences.

The reference style is not following the journal style.

[Respond] Thanks for your comments. We have revised the reference style following journal requirements.

The authors should have a simple summary and conflict of interest statement.

[Respond] We have supplemented the summary and conflict of interest statement after the manuscript.

The threshold for Bonferroni should be 0.05/N

[Respond] We refer to R.J.Simes (1986) ‘A improved Bonferroni procedure for multiple tests of significance’. In that paper, the Bonferroni threshold was defined as a/N, whatever a equals to 0.01 or 0.05. The a is just a confidence coefficient for multiple testing. So We believe that 0.01/N is more strict and suitable than 0.05/N.

Reviewer 3 Report

In this paper “Genome-Wide Association Mapping and Genetic Evaluation of Body Weight Traits in Yaks” by Wang et al. evaluated the population stratification, heritability of Mawa yak population, and identified candidate functional genes related to body weight using whole-genome sequencing. Overall, this manuscript is interesting to the field and well written.

How do authors treat the BW at different ages in GWAS analysis?

Line 75 Usually, Sonication was used to break genomic DNA into small DNA fragments, why do authors use restriction enzyme to cut the DNA.

It is recommended to add parameters for the software programs used to call SNP.

Line 28 Please change yak to yaks.

Line 244-245 This sentence is not accurate.

Line 279-280 This sentence is no sense.

Author Response

In this paper “Genome-Wide Association Mapping and Genetic Evaluation of Body Weight Traits in Yaks” by Wang et al. evaluated the population stratification, heritability of Mawa yak population, and identified candidate functional genes related to body weight using whole-genome sequencing. Overall, this manuscript is interesting to the field and well written.

[Respond] Thanks for sharing your time on our manuscript.

How do authors treat the BW at different ages in GWAS analysis?

[Respond] Thanks for your comments. We treat ages as covariance in the GWAS model. We have supplemented details in the formula and sentences. Supplement ages in the formula and sentences.

Line 75 Usually, Sonication was used to break genomic DNA into small DNA fragments, why do authors use restriction enzyme to cut the DNA.

[Respond] Thanks for your comments. Sonication breaking usually makes small DNA fragments with random breaking. The random breaking needs more sequencing depth to make sure the accuracy of sequencing. The restriction enzyme breaking is better to approach assembly fragments, although there are fewer markers detected by restriction enzyme breaking than sonication breaking.

It is recommended to add parameters for the software programs used to call SNP.

[Respond] Thanks for your comments. We have added relative parameters and codes for calling SNP.

Line 28 Please change “yak” to “yaks”.

[Respond] We agree with you and have changed the words.

Line 244-245 This sentence is not accurate.

[Respond] Thanks for your comments. We have revised this sentence.

Line 279-280 This sentence is no sense.

[Respond] Thanks for your comments. We have removed this sentence.

Round 2

Reviewer 1 Report

Congratulation!

Clear and well done!

Reviewer 2 Report

The authors have clarified my concerns about the models and confounding effects I have some minor suggestions.

Please rewire the simple summary, the flow of information is not good, and some terms are not clear.

Line 12: Change Tibetan to Tibetan people.

Genetic potency ability: it is not a clear term, that might change to genetic merit.

Location of candidate genes is not right, change to causal genes or mutations.

The sentence “To reveal these relationships, it will help us to understand the genetic rule of large mammal growth and development” is not clear, what the authors mean by these relationship. What is “it” in the second phrase? Please rewrite it.

“Accuracy detection of candidate genes” might change to Identified potential candidate genes

Remove a dot (.) in line 92 a

Line 295: Add the table title here.

Gene names should be in Italic

Author Response

The authors have clarified my concerns about the models and confounding effects I have some minor suggestions.

Please rewire the simple summary, the flow of information is not good, and some terms are not clear.

[Respond] Thanks for your comments. We have followed your comments and revised our manuscript.

Line 12: Change Tibetan to Tibetan people.

[Respond] Yes, that word should be added.

Genetic potency ability: it is not a clear term, that might change to genetic merit.

[Respond] Thanks for your comments. We think “genetic merit” should be genetic variants, such as SNP, Indel, or CNV. But in this sentence, we want to talk about heritability, detection of causal genes, and breeding values. So we revised this sentence as “The estimated genetic parameters such heritability, detection of candidate genes, and estimated breeding values are important to evaluate individual genetic potency ability”.

Location of candidate genes is not right, change to causal genes or mutations.

[Respond] We have revised this whole sentence.

The sentence “To reveal these relationships, it will help us to understand the genetic rule of large mammal growth and development” is not clear, what the authors mean by these relationship. What is “it” in the second phrase? Please rewrite it.

[Respond] We agree with your comment and have revised this whole sentence.

“Accuracy detection of candidate genes” might change to Identified potential candidate genes

[Respond] Revised.

Remove a dot (.) in line 92 a

[Respond] Revised.

Line 295: Add the table title here.

[Respond] Thanks for your comment. The above information about the body size of Jiulong yaks is referred to relative papers. Here we just used these values to show the biggest body size yaks population. We have no other values from other breeds. So we decide to only use text to describe this background not as a table.

Gene names should be in Italic

[Respond] We have made sure all gene names are in Italic style.

Reviewer 3 Report

The authors have revised MS following my suggestion.

Author Response

Thanks for your comments. These will help us to improve our manuscript easy to understand.

Round 3

Reviewer 2 Report

The authors responded to my comments, but I was not satisfied with their corrections.  Please pay attention to the simple summary as it is the first important message of the manuscript. 

The estimated genetic parameters such as heritability, detection of causal genes, and estimated breeding values are essential to evaluate individual genetic potency ability

Firstly, estimated genetic parameters do not include the detection of causal variants and estimated breeding values.

Secondly, I suggest changing causal genes to causal variants is better.

Line 16: What does it mean “From whole-genome view”, what the involvement of the genetic parameters and whole-genome view.

Line 17: Again, the authors should not use genetic parameters

The authors should use simple text for a simple summary. 

Line 159:  After this program should change to After removing outliners

Author Response

The authors responded to my comments, but I was not satisfied with their corrections.  Please pay attention to the simple summary as it is the first important message of the manuscript. 

[Respond] Thanks for your comments. We have revised the simple summary carefully following your comments.

The estimated genetic parameters such as heritability, detection of causal genes, and estimated breeding values are essential to evaluate individual genetic potency ability

Firstly, estimated genetic parameters do not include the detection of causal variants and estimated breeding values.

[Respond] We agree with you. We have re-write this sentence.

Secondly, I suggest changing causal genes to causal variants is better.

[Respond] Yes, the variants contain multiple causal types. This word should be appropriate.

Line 16: What does it mean “From whole-genome view”, what the involvement of the genetic parameters and whole-genome view.

[Respond] Thanks for your comments. We have re-write this sentence.

Line 17: Again, the authors should not use genetic parameters

 [Respond] We have used other words replaced.

The authors should use simple text for a simple summary. 

[Respond] We have rewritten this section and avoided using complex words.

Line 159:  After this program should change to After removing outliners

[Respond] Revised.

Round 4

Reviewer 2 Report

Dear authors, 

Thank you for your responses. 

I suggest the author remove the term "genetic evaluation" in the manuscript title as the authors did not perform any research related to it.

Change Genome-wide association mapping to genome-wide association study as well.

For a simple summary, the authors might write some text similar to it "Yak is the largest mammal in the Qinghai-Tibetan Plateau and is an important food and financial resource for Tibetan people. Body weight is a valuable trait for this species. In this study, we performed a genome-wide association study using the whole genome sequencing data to detect the potential candidate genes for body weight. We identified several potential candidate genes for the trait such as MFSD4, LRRC37B, and NCAM2.  The results of the study are important for the development of breeding programs for improving body weight in Yak." 

I hope the authors put more effort to make a good simple summary. 

Author Response

I suggest the author remove the term "genetic evaluation" in the manuscript title as the authors did not perform any research related to it.

[respond] Thanks for your comments. We have checked the definition of “genetic evaluation”, which does not match our manuscript. We have removed this word in the title and the whole manuscript.

Change Genome-wide association mapping to genome-wide association study as well.

[respond] We agree with your comment. It has been revised.

For a simple summary, the authors might write some text similar to it "Yak is the largest mammal in the Qinghai-Tibetan Plateau and is an important food and financial resource for Tibetan people. Body weight is a valuable trait for this species. In this study, we performed a genome-wide association study using the whole genome sequencing data to detect the potential candidate genes for body weight. We identified several potential candidate genes for the trait such as MFSD4, LRRC37B, and NCAM2.  The results of the study are important for the development of breeding programs for improving body weight in Yak." 

[respond] Thanks for your example. That is very useful for our manuscript. We noticed that “no abbreviations” is required by the animals-template file from this journal. So we replaced these genes' names with “functional genes”.

I hope the authors put more effort to make a good simple summary. 

[respond]Thanks for your patient comments. We have re-written the simple summary.

Round 5

Reviewer 2 Report

The authors have addressed my comments. Just small suggestion to remove "for development"  in line 17.